# GS9 acts as a transcriptional activator to regulate rice grain shape and appearance quality

Dong-Sheng Zhao[1], Qian-Feng Li[1,2], Chang-Quan Zhang[1,2], Chen Zhang[1], Qing-Qing Yang[1], Li-Xu Pan[1], Xin-Yu Ren[1], Jun Lu[1], Ming-Hong Gu[1] & Qiao-Quan Liu [1,2]

Identification of grain shape determining genes can facilitate breeding of rice cultivars with optimal grain shape and appearance quality. Here, we identify *GS9* (*Grain Shape Gene on Chromosome 9*) gene by map-based cloning. The *gs9* null mutant has slender grains, while overexpression *GS9* results in round grains. *GS9* encodes a protein without known conserved functional domain. It regulates grain shape by altering cell division. The interaction of GS9 and ovate family proteins OsOFP14 and OsOFP8 is modulated by OsGSK2 kinase, a key regulator of the brassinosteroids signaling pathway. Genetic interaction analysis reveals that *GS9* functions independently from other previously identified grain size genes. Introducing the *gs9* allele into elite rice cultivars significantly improves grain shape and appearance quality. It suggests potential application of *gs9*, alone or in combination with other grain size determining genes, in breeding of rice varieties with optimized grain shape.

---

[1] Key Laboratory of Plant Functional Genomics of the Ministry of Education/Key Laboratory of Crop Genetics and Physiology of Jiangsu Province, College of Agriculture, Yangzhou University, 225009 Yangzhou, China. [2] Co-Innovation Center for Modern Production Technology of Grain Crops of Jiangsu Province/ Joint International Research Laboratory of Agriculture and Agri-Product Safety of the Ministry of Education, Yangzhou University, 225009 Yangzhou, China. These authors contributed equally: Dong-Sheng Zhao, Qian-Feng Li and Chang-Quan Zhang. Correspondence and requests for materials should be addressed to Q.-Q.L. (email: qqliu@yzu.edu.cn)

In rice, grain morphology, notably grain size or shape (GS), is one of the key determinants of both yield and quality. In general, large grain size is associated with high yield but poor quality, especially the appearance of milled rice, such as chalkiness[1, 2]. An important goal of rice breeding programs is therefore a high yield plus ideal grain shape. Slender rice grains are highly commercial worldwide due to their transparent appearance and lack of an undesirable chalky texture and taste[3].

To date, several genes and major quantitative trait loci (QTLs) controlling grain morphology have been cloned and characterized, most of which play a role in regulating grain size and yield, with counteractive effects on grain appearance[4–20]. These cloned genes have provided considerable insight into the individual molecular basis of grain size and shape regulation[6, 21]. For example, GS3 is known to encode a putative trans-membrane protein involved in the G protein signal pathway[7], while GW2 encodes a RING-type protein with E3 ubiquitin ligase activity[11]. Furthermore, qSW5/GW5 encodes a polyubiquitin-binding protein involved in the ubiquitin-proteasome pathway[12, 22]. Transcriptional regulatory factors also play important roles in controlling grain size[6]. GLW7 encodes the plant-specific transcription factor OsSPL13, which positively regulates cell size[10], while GL7 encodes Arabidopsis homologous protein LONGIFOLIA[20]. OsSPL16/GW8 encodes a SBP-domain transcription factor that binds directly to the GW7/GL7 promoter and represses expression[14, 19], while the OsMKK4-OsMAPK5 regulatory module plays an important role in regulating grain size by controlling cell proliferation in spikelet hulls[6, 23, 24]. Plant hormones are also directly or indirectly involved in controlling grain size. qGL3 encodes the protein phosphatase OsPPKL1, which might participate in brassinosteroids (BR) signaling in rice[6, 9], and GL3.1 controls grain size by regulating Cyclin-T1-3[8]. Moreover, several other genes, including GS5, GS2/GL2, and GW5, are also involved in BR signaling[6, 13, 15–17, 25]. TGW6 and BG1, which encode an IAA-glucose hydrolase activity protein and membrane localized protein, respectively, are thought to be specifically induced by auxin[18, 26].

Although several important genes involved in the regulation of grain size and shape have been cloned in rice, their regulation mechanism and network remain largely unknown[6, 21]. Functional characterization of additional genes/QTLs involved in determining grain shape is therefore important in furthering our understanding of the molecular mechanisms behind regulation of grain traits[5], as well as helping meet the demand for high yield and quality cultivars[1, 27–29].

From a series of chromosome segment substitutional lines (CSSLs) derived from two subspecies[30], we identify the CSSL N138, which has a slender grain shape but identical grain weight as its recipient Nipponbare (NPB) (Supplementary Fig. 1). Its target gene, GS9 (Grain Shape Gene on Chromosome 9), is found to encode a transcriptional activator involved in regulation of grain shape. Furthermore, introducing gs9 null allele produces a slender grain with excellent appearance quality. These findings may facilitate our understanding of the mechanism of grain shape in rice.

## Results

**Slender grain shape in N138 line is controlled by gs9.** The CSSL N138 harbors six homozygous chromosomal segments substituted from the donor Qingluzan11 (indica) in Nipponbare background (Supplementary Fig. 1b). It exhibited a more slender grain phenotype than that of Nipponbare. (Fig. 1a–d), and no obvious difference was observed in grain thickness or any other agronomic traits.

Mature seeds of F₁ plants derived from N138 and Nipponbare exhibited a similar morphology as Nipponbare (Supplementary Fig. 2a). Among the derived F₂ population, the ratio of

normal (Nipponbare) to slender (N138) grains was around 3:1 (Supplementary Fig. 2b). Linkage analysis revealed that the slender phenotype was tightly co-segregated with the substitutional segment of 5.9 Mb long on chromosome 9 (Fig. 1e and Supplementary Fig. 1b). These findings suggest that the slender grain phenotype in N138 is controlled by a single recessive gene within this substitutional segment. The alleles in N138 and Nipponbare were therefore named gs9 and GS9, respectively.

**Map-based cloning of GS9.** The GS9 locus was initially mapped to a region between markers RM6235 and IN0913 using the above F₂ population (Fig. 1e), and further narrowed to a 5.9-kb genomic DNA region between markers IN0927 and IN0929 using F₃ population (Fig. 1f, g). Only one candidate gene, LOC_Os09g27590, was subsequently identified as having the predicted open reading frame (ORF) (Fig. 1h and Supplementary Fig. 1e), hereafter referred to as GS9.

Based on the rice annotation database, LOC_Os09g27590/GS9 contains four exons and three introns (Fig. 1h). Sequencing data showed six polymorphic sequences in the coding region of GS9 between N138 and Nipponbare (Fig. 1h), including one 3-bp (GCG) insertion in the first exon (InDel1), four single nucleotide polymorphisms (SNPs) in the fourth exon (SNP1–SNP4), and one 7-kb insertion in the second exon (InDel2) (Fig. 1h, Supplementary Fig. 3). Except for InDel2, the remaining five polymorphisms also existed between the two parents, Nipponbare and Qingluzan11, resulting in the insertion of one amino-acid residue and substitution of three residues (Fig. 1h). The 7-kb insertion of InDel2 was subsequently amplified from the N138 genome, revealing a relatively complex sequence without the predicted ORF. Partial repeats were observed at both the 5′ and 3′ ends, matching multiple regions in the rice genome thought to have originated from recombination of various repeated sequences (Supplementary Fig. 3).

Two more CSSLs, N107 and N139, were also found to carry the substituted segment covering the GS9 locus from 9311 and Qingluzan11, respectively (Supplementary Fig. 1c, d). The LOC_Os09g27590/GS9 locus in N107 and N139 was identical to that in 9311 and Qingluzan11 (Fig. 1h). Interestingly, both lines displayed a normal grain phenotype like Nipponbare (Fig. 1h and Supplementary Fig. 1a–d). Additional linkage analysis of the F₂ population revealed that the slender grain phenotype was tightly co-segregated with the 7-kb insertion (Supplemental Fig. 3d, e). These findings suggest that LOC_Os09g27590 is the candidate gene for GS9, and the 7-kb insertion resulted in the loss-of-function and subsequent slender grain phenotype in N138.

**Confirmation of the GS9 gene.** The near-isogenic line (NIL) NIL-gs9 containing gs9 allele within the Nipponbare background was developed for complementary and functional tests. NIL-gs9 and Nipponbare showed identical growth, development, and final phenotype (Fig. 2a, b). Mature grains of NIL-gs9 were significantly more slender, causing about 24% increase in the ratio of grain length to width (Fig. 2c–e).

As expected, the spikelet hulls of the complementary plants NIL-gs9-C perfectly resembled those of Nipponbare (Supplementary Fig. 4a–f), without any obvious morphological differences (Supplementary Fig. 4b). Mature grains of NIL-gs9-C also resembled those of Nipponbare, being rounder than those of NIL-gs9 (Fig. 2h–k). One homologous mutant NPB-cas9-1, produced in Nipponbare background using CRISPR/cas9 approach, had a 1-bp insertion and subsequent frameshift mutation in the coding region of LOC_Os09g27590 (Supplementary Fig. 5a). The NPB-cas9-1 mutant grew normally,

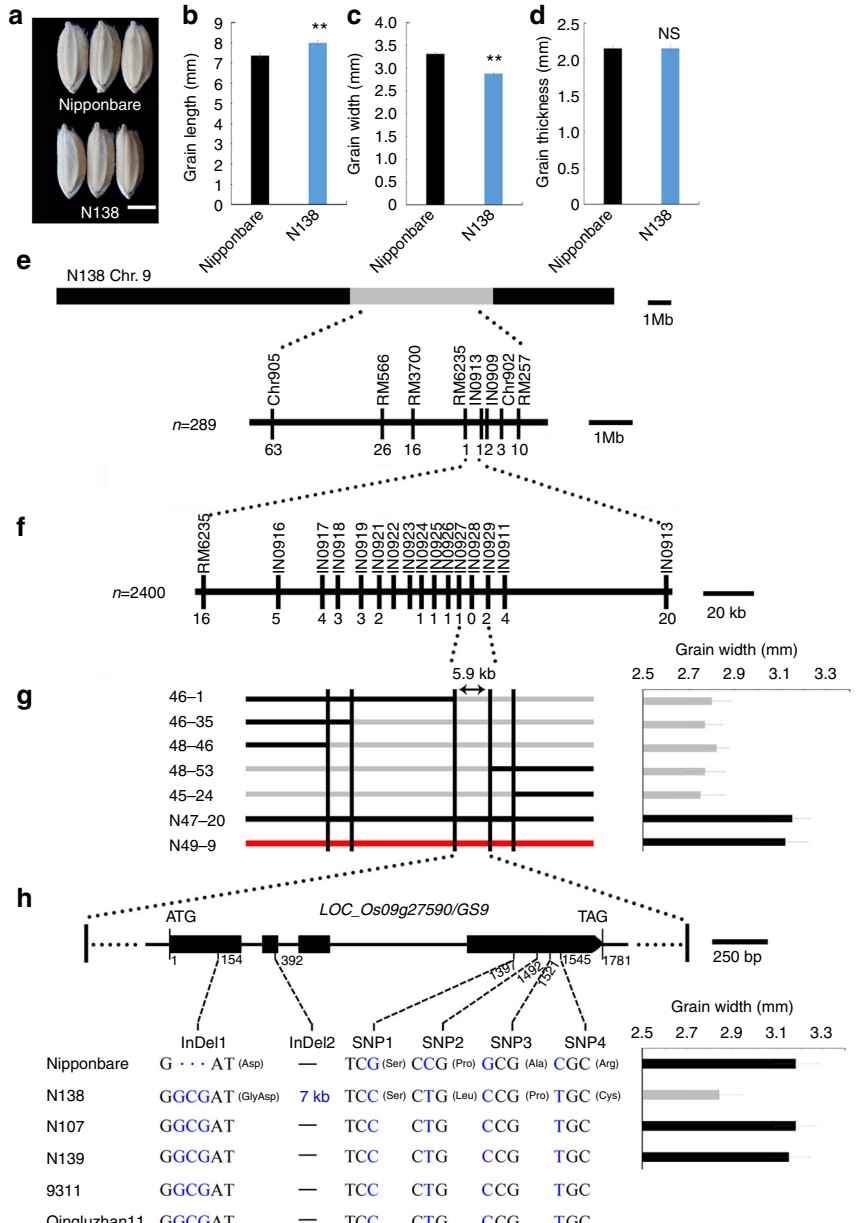

**Fig. 1** Map-based cloning of *GS9*. The morphology (**a**), length (**b**), width (**c**), and thickness (**d**) of mature grains from CSSL line N138 and its recipient Nipponbare. Scale bar, 3 mm. Data are given as means ± standard deviation (SD, *n* = 30); ** significant difference (*P* < 0.01, *t*-test). NS not significant difference. **e** *GS9* was preliminary mapped between markers RM6235 and IN0913 on chromosome 9 using 289 F₂ recessive individuals from the cross of N138 and Nipponbare. **f** *GS9* was narrowed down to a 5.9-kb genomic DNA region between markers IN0927 and IN0929 using 2400 F₃ recessive individuals. The numbers below the bar in panels **e** and **f** indicate the number of recombinants between *GS9* and the molecular markers shown. **g** Genotype (left) and grain width (right) were shown for recombinant plants and the control plants. Black and gray bars represent chromosomal segments homozygous for Nipponbare and N138 alleles, respectively, while red one indicates heterozygous. **h** Schematic representation of the structure and allelic variation of the candidate gene *LOC_Os09g27590 (GS9)*, the only one predicted open reading frame in the 5.9-kb delimited region, among three CSSLs and their parents. The numbers below the gene bar indicate the number of polymorphic nucleotide(s) located in coding region, where the first nucleotide of start codon is as No. 1. Differential nucleotides are shown in blue and resulting amino acids are in brackets. The grain width of three CSSLs and their recipient Nipponbare is shown on the right

(Supplementary Fig. 5b–e), but presented a more slender grain as that of NIL-*gs9* (Fig. 2l–o and Supplementary Fig. 5f–i).

The *GS9* overexpression lines, NPB-OE and NIL-*gs9*-OE, were generated in Nipponbare and NIL-*gs9* background, respectively, and their mature grains were rounder than those of Nipponbare (Fig. 2l–o and Supplementary Fig. 6a–f). The ectopic expression of *GS9* also caused other phenotypic changes such as a slight decrease in plant height and panicle length, and a shortened leaf (Supplementary Fig. 6g–m). Overall, these results verified that

*LOC_Os09g27590* is the target gene of *GS9*, acting as a dominant and specific regulator of grain shape in rice.

**GS9 regulates grain shape by altering cell division.** Dynamic observations during grain filling showed that the caryopsis of NIL-*gs9* was significantly more slender than that of Nipponbare from 7 days after fertilization (DAF) (Supplementary Fig. 7d). Interestingly, there was no difference in caryopsis size between

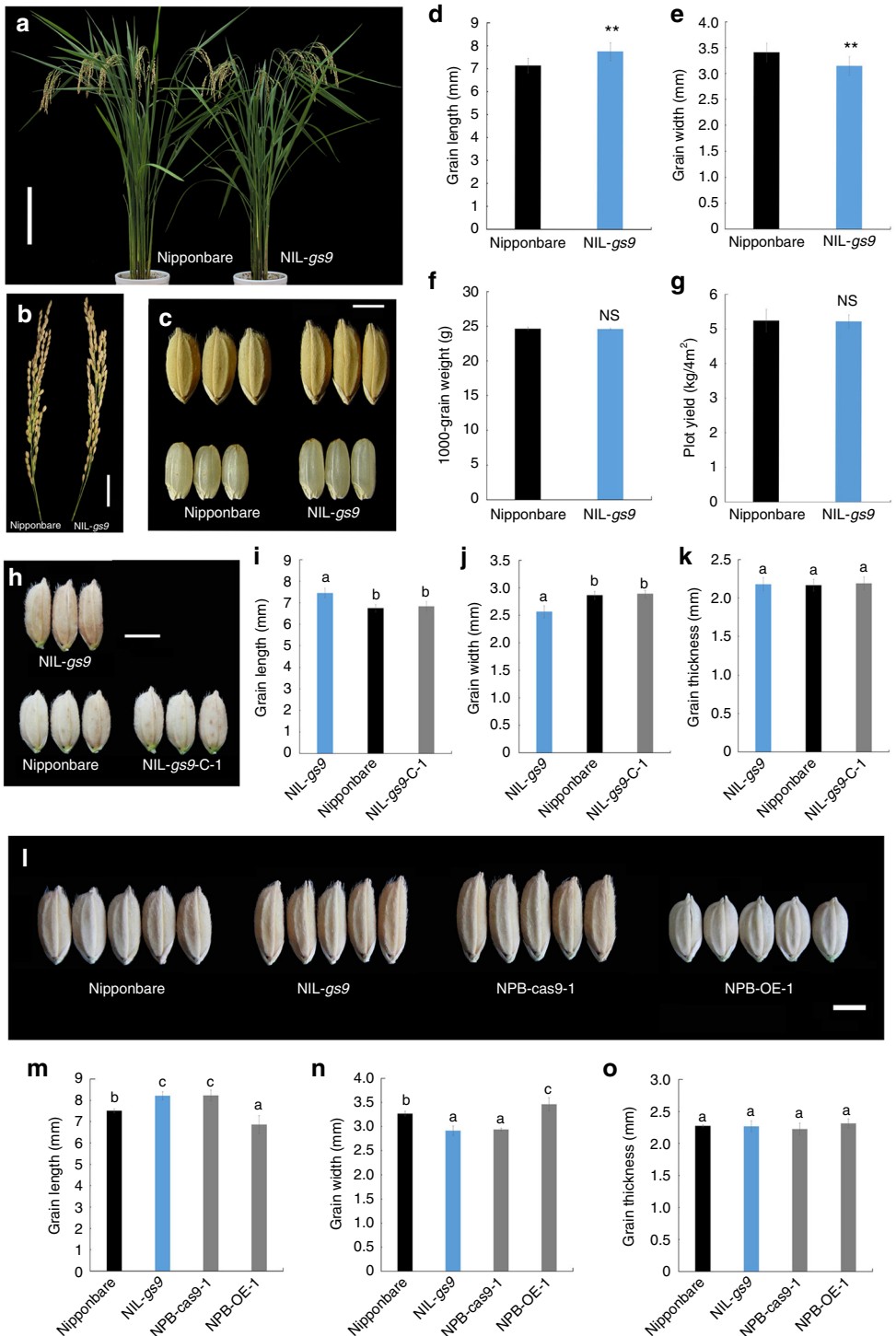

**Fig. 2** Confirmation of *GS9* in regulating grain shape. **a**–**g** Confirmation by using near-isogenic line. Morphology of whole plant (**a**), panicle (**b**), grain shape (**c**–**e**) and weight (**f**), and yield (**g**) of the near-isogenic line NIL-*gs9* and Nipponbare during maturation from the same planting condition in the summer of 2014. Scale bar, 20 cm (**a**), 3 cm (**b**), 3 mm (**c**). Data in d and e are given as means ± SD ($n = 30$); ** significant difference ($P < 0.01$, *t*-test). NS not significant difference (*t*-test). **h**-**k** Complementary test. Grain shape (**h**), length (**i**), width (**j**), and thickness (**k**) of NIL-*gs9* and its complementary transgenic line NIL-*gs9*-C, and their parent Nipponbare from the same planting condition in 2015. Scale bar, 3 mm. Data are given as means ± SD ($n = 30$). **l**-**o**, CRISPR/Cas9 editing and overexpression of *GS9* in Nipponbare. Comparison of grain shape (**i**), length (**m**), width (**n**), and thickness (**o**) among different lines from the same planting condition in 2016. NPB-cas9-1, a null *gs9* mutant in Nipponbare background created by CRISPR/Cas9 editing system; NPB-OE-1, a *GS9* overexpression line in Nipponbare background. Scale bar, 3 mm. Data in **m**-**o** are given as means ± SD ($n = 20$). The presence of the different lowercase letters in **i**-**k** and **m**-**o** denotes significant difference between the means ($P < 0.01$, one-way ANOVA)

Nipponbare and NIL-*gs9* from 1 to 5 DAF (Supplementary Fig. 7d), suggesting that the slender mature grains in NIL-*gs9* were the result of differentiation of the spikelet hull (Supplementary Fig. 7a–c).

As expected, the spikelet hull just before fertilization was also much slender in NIL-*gs9* than Nipponbare (Fig. 3a and Supplementary Fig. 7a–c). Consistently, examination of a cross-section of the spikelet hull showed a significantly shorter spikelet perimeter, and decreased number of lower epidermis cells, both palea and lemma, in NIL-*gs9* (Fig. 3b–f). It suggests that cell division was significantly reduced in a transverse direction in the NIL-*gs9* spikelet hulls. However, an obvious increase in cell size was observed in NIL-*gs9*, possibly due to compensation[31]. Although the spikelet perimeter decreased in NIL-*gs9*, the number of rows of specialized cells with a rigid wall in the upper epidermis was unchanged (Fig. 3b, c). In addition, there is no

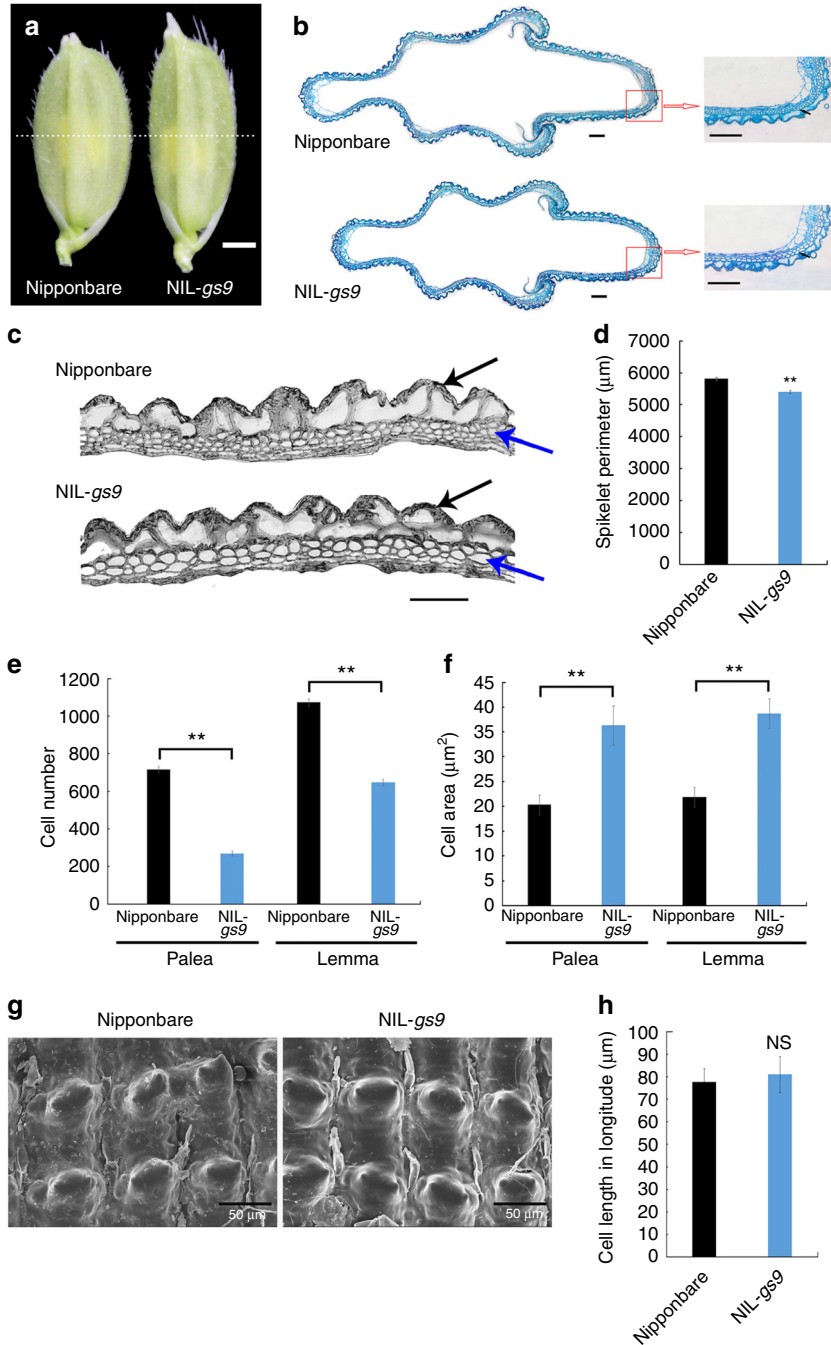

**Fig. 3** Histological comparison of the spikelet hulls between NIL-*gs9* and Nipponbare. **a** Spikelet hulls before anthesis. The white dashed line indicates the sites of the cross-sections shown in **b**. Scale bar, 1 mm. **b** Cross-sections of spikelet hulls. The right-hand images show close-up views of the boxed region. Scale bar, 100 μm. The black arrows indicate the lower epidermis cells. **c** Magnified views of the cross-sections indicated in **b**. The black and blue arrows indicate the rows of specialized cells with rigid walls and the lower epidermis cells, respectively. Scale bar, 50 μm. Spikelet perimeter (**d**), cell number (**e**), and cell area (**f**) of palea and lemma. Data are given in means ± SD, with three biological replicates. ** significant difference (*P* < 0.01, *t*-test). **g** Scanning electron microscopy of the outer surfaces of the glumes. Scale bar, 50 μm. **h** The cell length in longitude. Data are given as means ± SD; *n* = 15. NS not significant difference (*t*-test)

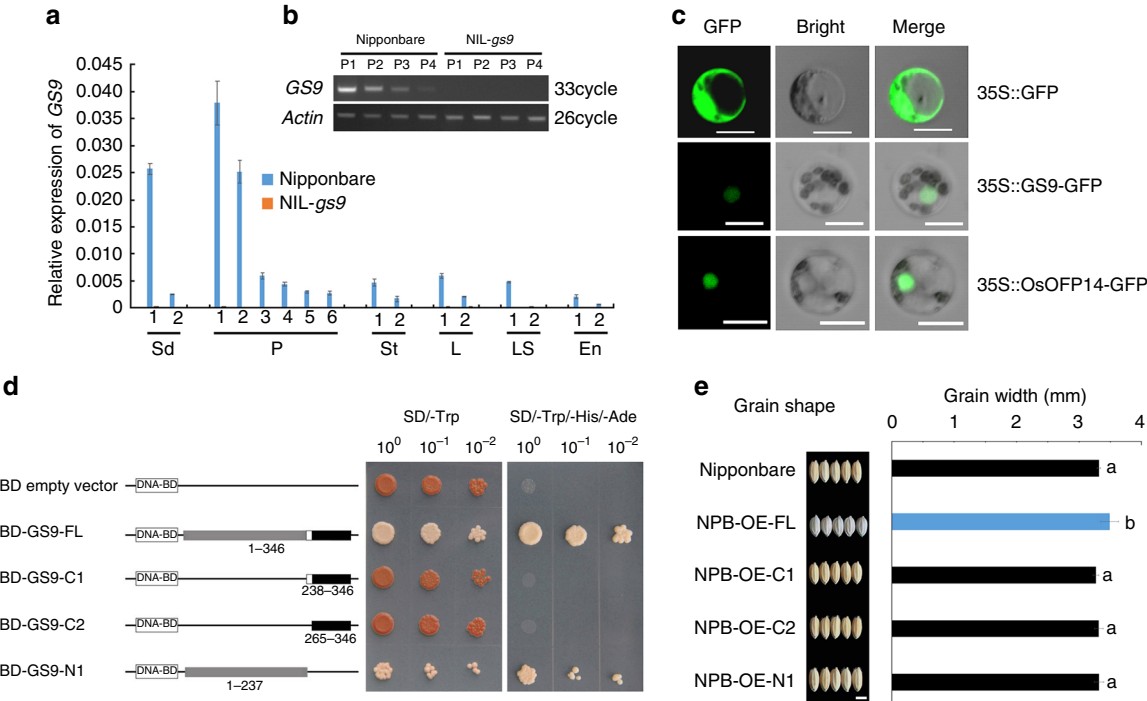

**Fig. 4** GS9 acts as a transcriptional activator. **a** GS9 transcript levels in different organs. Sd1 and Sd2, leaf and roots of seedlings, respectively; P1–P4, young panicles with the average lengths of about 2 cm, 5 cm, 10 cm, and >10 cm, respectively; while P5–P6, panicles before heading or 4 days after heading, respectively; St1, L1, and LS1, the stem, leaf, and leaf sheath at panicle differentiation, respectively; St2, L2, and LS2, the stem, leaf, and leaf sheath just before heading; En1 and En2, endosperms 5 and 25 days after fertilization, respectively. Rice *actin* gene was used as a control. Values are shown as means ± SD ($n = 3$). **b** No GS9 transcript was detected in NIL-gs9 shown by using semi-quantitative PCR. P1–P4, the same as in **a**. **c** Subcellular localization of GS9 (35 S::GS9-GFP) and OsOFP14 (35 S::OsOFP14-GFP) in rice protoplasts. 35S::GFP was the control containing only the GFP coding region. Scale bar, 10 μm. **d** Transcription activity assay of full-length or truncations of GS9 in yeast. BD-GS9-FL, BD-GS9-C1, BD-GS9-C2, and BD-GS9-N1 showed the GAL4 DNA-binding domain in pGBKT7 vector fused with full-length, the regions encoding C-terminal, or N-terminal of GS9, respectively. Numbers below the bar indicate the amino acid residues used for construction. **e** Shape and width of mature grains from overexpression plants and wild-type Nipponbare. NPB-OE-FL, NPB-OE-C1, NPB-OE-C2, and NPB-OE-N1 represented the transformants overexpressed of full-length or truncations of GS9 as shown in **d**. All were driven by the *CaMV 35S* promoter. Scale bar, 4 mm. Values are shown as means ± SD ($n = 30$). Different letters denote significant difference ($P < 0.01$, one-way ANOVA)

significant difference in longitudinal cell density on the outer surface of the glume between NIL-gs9 and Nipponbare (Fig. 3g, h). Thus, it suggested that the longer grain length of NIL-gs9 was due to an increase in longitudinal cells in the spikelet hull, while the narrower width was the result of a decrease in transverse cell number. We thus compared the expression of several genes involved in cell cycle, such as *CDKA1*, *H1*, *CYCB2.2*, *CYClaZm*, *CDKB*, and *MAD2*, and found that they were upregulated in young panicles of NIL-gs9 (Supplementary Fig. 8), suggesting that the altered cell number might result from elevated expression of genes promoting cell proliferation. Overall, it appears that the loss-of-function of *GS9* in NIL-gs9 altered cell division during spikelet development.

***GS9* encodes a transcriptional activator**. In Nipponbare, *GS9* transcript level was preferentially high in young panicles, decreasing gradually during panicle development (Fig. 4a, b). In transgenic rice containing the *GUS* (β-glucuronidase) reporter gene driven by *GS9* promoter, high GUS activity was detected in inflorescences and developing spikelets, with weak activity in developed spikelets (Supplementary Fig. 9). This expression pattern was consistent with the role of *GS9* in determining spikelet hull development. In NIL-gs9, no *GS9* transcript was detected in any of the tested organs or tissues (Fig. 4a, b).

*GS9* encodes an unknown expressed protein with a number of orthologs in higher plants; however, none has been functionally characterized and no known-functional domain has yet been predicted. Phylogenetic analysis classified these proteins into two subgroups, dicots and monocots (Supplementary Fig. 10). GS9-like proteins from rice and seven other Poaceae species formed a small clade with a high bootstrap value and strong similarity in the C-terminal region (265–346 residues) (Supplementary Fig. 10 and 11). In addition, no GS9 orthologous proteins were found in any other organism, suggesting that GS9 is a regulator specific to higher plants.

The GS9::GFP fusion protein localized in the nucleus of either rice protoplast (Fig. 4c) or tobacco epidermal cells (Supplementary Fig. 12). Furthermore, transcription activation assay in yeast showed transcriptional activation activity of the full-length *GS9* (BD-GS9-FL) (Fig. 4d). Data from dual-luciferase assays further confirmed that *GS9* functions as a transcription activator (Fig. 5d). Interestingly, the N-terminal region of the GS9 protein (1–237 residues), BD-GS9-N1, was found to be sufficient for activation of the reporter, while its C-terminal truncates, BD-GS9-C1 (238–346 residues) and BD-GS9-C2 (265–346 residues), were not (Fig. 4d). These data suggest that the N- and C-terminals of GS9 are involved in transcription activation and DNA-binding activity, respectively.

Accordingly, four types of transgenic plants with overexpression of different GS9 domains were generated, designated as

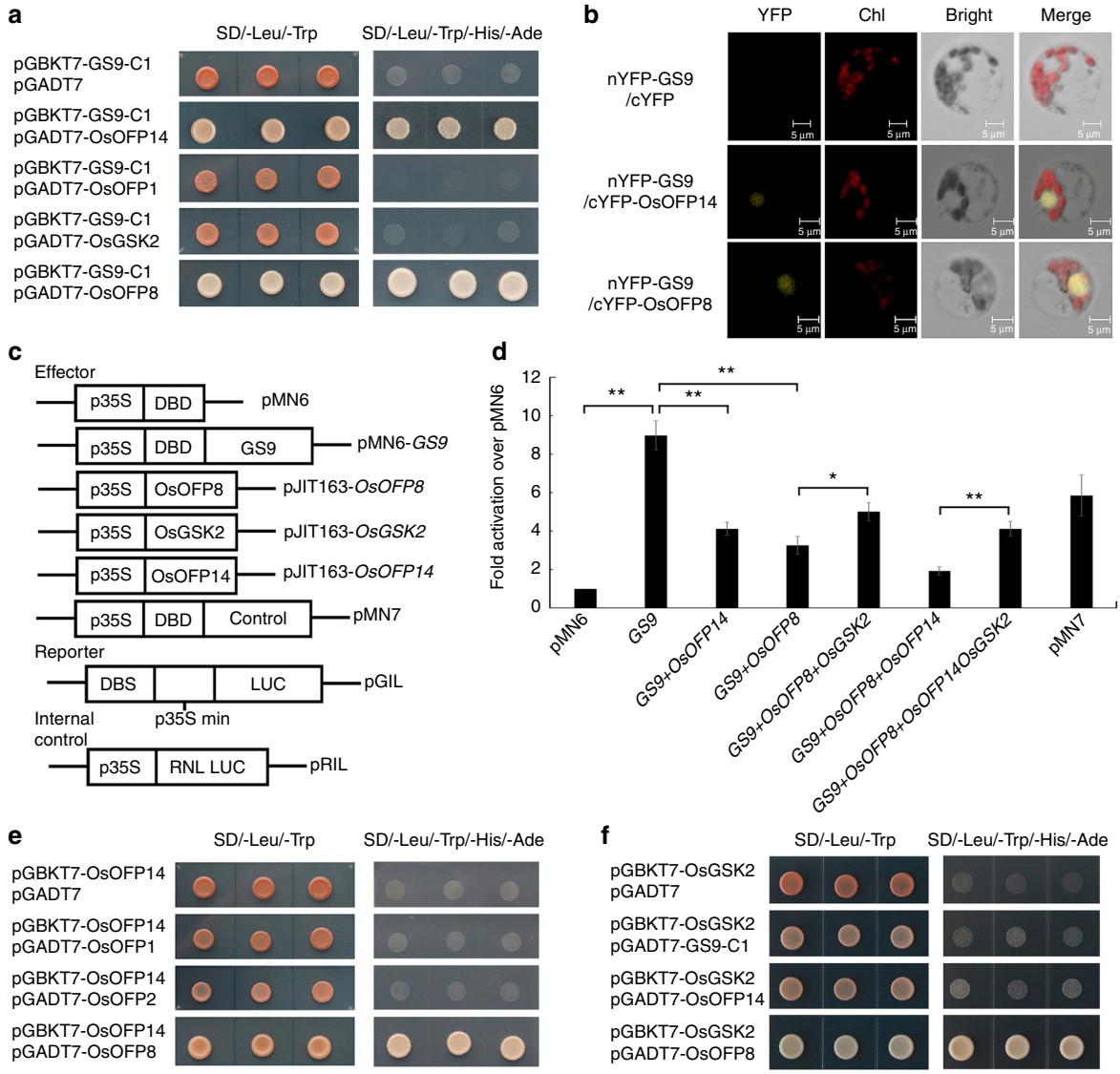

**Fig. 5** OsOFP14 interacts with GS9 and decreases its transcription activity. **a** Analyses of interactions between GS9 and OsOFP14, OsOFP1, OsOFP8, OsGSK2 in yeast two-hybrid assay, respectively. GS9-C1 means the vector containing the C1 terminal of GS9 as shown in Fig. 4d. OsOFP14, LOC_Os04g33870; OsOFP1, LOC_Os01g12690; OsOFP8, LOC_Os01g64430; OsGSK2, LOC_Os05g11730. **b** Confirmation of the interaction between GS9 and OsOFP14, OsOFP8 in rice protoplasts using BiFC assay. nYFP and cYFP, N-terminal and C-terminal of YFP, respectively. Chl chlorophyll. Scale bar, 50 μm. **c, d** Transcription activity assay using the pMN6 system. The main structure of vectors is shown in **c**. LUC firefly luciferase; RNL LUC renilla luciferase. **d** shows the transcription activity in *Arabidopsis* protoplasts by co-transformation of different effector vector(s) with the reporter plasmid pGIL and internal control pRIL. Data are given as means ± SD, with three biological replicates. * or ** significant difference (P < 0.05 or 0.01, t-test). **e, f** Analyses of interactions among OsOFP14, OsOFP8, OsOFP1, and OsGSK2 in yeast two-hybrid assay

NPB-OE-FL, NPB-OE-C1, NPB-OE-C2, and NPB-OE-N1, respectively (Supplementary Fig. 13a–f). However, grain shape alteration was only observed in the NPB-OE-FL transgenic plants containing the full-length CDS of *GS9* (Fig. 4e and Supplementary Fig. 13c, g–i). Taken together, these results suggest that full-length GS9 is required for the regulation of grain shape in rice.

**GS9 interacts with OsOFP14 and OsOFP8.** To determine the underlying mechanism of GS9 in regulating grain shape, 11 potential GS9-interacting proteins (GIPs) were screened using a yeast two-hybrid (Y2H) assay (Supplementary Table 2), including OsOFP14/LOC_Os04g33870, an OVATE family protein (OFP), which usually functions in fruit shape[32]. The interaction between GS9 and OsOFP14 was further confirmed using full-length cDNA

of *OsOFP14* (Fig. 5a). Interestingly, OsOFP14 was also localized to the nucleus, as with GS9 (Fig. 4c and Supplementary Fig. 12). Moreover, bimolecular fluorescence complementation (BiFC) analyses in both epidermal and protoplast cells further confirmed in vivo interaction between GS9 and OsOFP14 in the nucleus of plant cells (Fig. 5b and Supplementary Fig. 14). Since OsOFP14 is a member of a transcription suppressor family[33], we speculated that OsOFP14 antagonistically or synergistically modulates GS9 activity. The dual-luciferase assays revealed the repression effect of OsOFP14 on GS9 transcriptional activity (Fig. 5c, d), confirming the role of OsOFP14 as a negative regulator of GS9. This modulated activity also implied that the GS9-OFP14 interaction indeed occurred in vivo.

There are several OFP members in rice; therefore, two additional representatives, OsOFP1/LOC_Os01g12690 and

OsOFP8/LOC_Os01g64430, were selected for further interaction analysis. The results supported the interaction between OsOFP8, but not OsOFP1, with either GS9 or OsOFP14 (Fig. 5a, e, f). Recently, OsOFP8 was identified as a regulator of rice development and growth via interaction with OsGSK2/LOC_Os05g11730, a key negative regulator of BR signaling pathway[34, 35]. Surprisingly, no interaction was detected between OsGSK2 and either GS9 or OsOFP14 (Fig. 5a, e, f). Thus, these findings suggest that GS9 is able to interact with OsOFP8 and OsOFP14, while OsGSK2 only interacts and phosphorylates OsOFP8 as reported[34]. The dual-luciferase assays also revealed the repression effect of OsOFP8 on the transcriptional activity of GS9 (Fig. 5c, d), and this repression could be partly recovered by OsGSK2 (Fig. 5d and Supplementary Fig. 15). As expected, co-expression of OsOFP14 and OsOFP8 had a more serious repression effect on GS9 transcriptional activity, which could also be attenuated by OsGSK2 (Fig. 5d). Overall, these findings suggest that at least three members, GS9, OsOFP14, and OsOFP8, are involved in transcription co-regulation, activity of which seems to be further modulated by OsGSK2 via the bridge of OsOFP8.

With the end to clarify whether OsOFP8 and OsGSK2 also regulate grain size, we obtained their transgenic rice lines[34, 35]. Interestingly, the OsOFP8-overexpression line (OsOFP8-OE7)[34] exhibited a more slender grain than that of the wild-type (Supplementary Fig. 16a–c), while the OsGSK2 RNA interference (RNAi) line (OsGSK2-Gi-2)[35] showed a 22% increase in grain length (Supplementary Fig. 17a–c). No difference was observed in longitudinal cell density on the outer surface of the glume between OsOFP8-OE7 or OsGSK2-Gi-2 and the wild types (Supplementary Fig. 16d, 17d). Thus, it suggested that the longer grain length of OsOFP8-overexpression or OsGSK2-RNAi lines was the result from an increase in longitudinal cell numbers of their spikelet hulls. Overall, these results further indicated that the

GS9-OsOFP8-OsGSK2 complex participates in regulation of grain shape, at least in part, by modulating cell division.

**Effects of *gs9* mutation on transcriptome.** To further explore the effects of *gs9* mutation on gene expression, we performed RNA-sequencing analysis with young panicles from NIL-*gs9* and Nipponbare. The data showed that the *gs9* null mutation resulted in a number of differentially expressed genes (DEGs) (Supplementary Data 1). Gene Ontology (GO) assay showed that these DEGs were significantly ($P < 0.05$) enriched for cellular component, molecular function, and biological process, including DNA binding activity and cell differentiation (Supplementary Fig. 18). Besides, the *gs9* mutation also caused the altered expression of some putative transcripts which preferred to express in rice inflorescence and glumes (http://ricexpro.dna.affrc.go.jp), such as LOC_Os01g46169, LOC_Os02g42950, and LOC_Os09g28370 (Supplementary Data 1), which was consistent with the fact that *gs9* null mutation altered cell division and inflorescence development.

As shown above, BR signaling was involved in GS9-mediated regulation of grain shape. However, our RNA-sequencing data revealed no significant difference between NIL-*gs9* and Nipponbare in the expression of genes involved in BR signaling (Supplementary Data 2). More interestingly, although OsOFP8 and OsOFP14 physically interacted with GS9 to suppress its transcriptional activity, the transcription of OFP genes was almost not altered in response to GS9 mutation (Supplementary Data 2). In addition, our quantitative RT-PCR data further verified the identical transcription of OsGSK2 and two OFP genes in NIL-*gs9* (Supplementary Fig. 19). All these data implied that the crosstalk among GS9 and OFPs as well as BR signaling pathway might not occur in the transcriptional level. However, the detailed mechanism still needs further clarification.

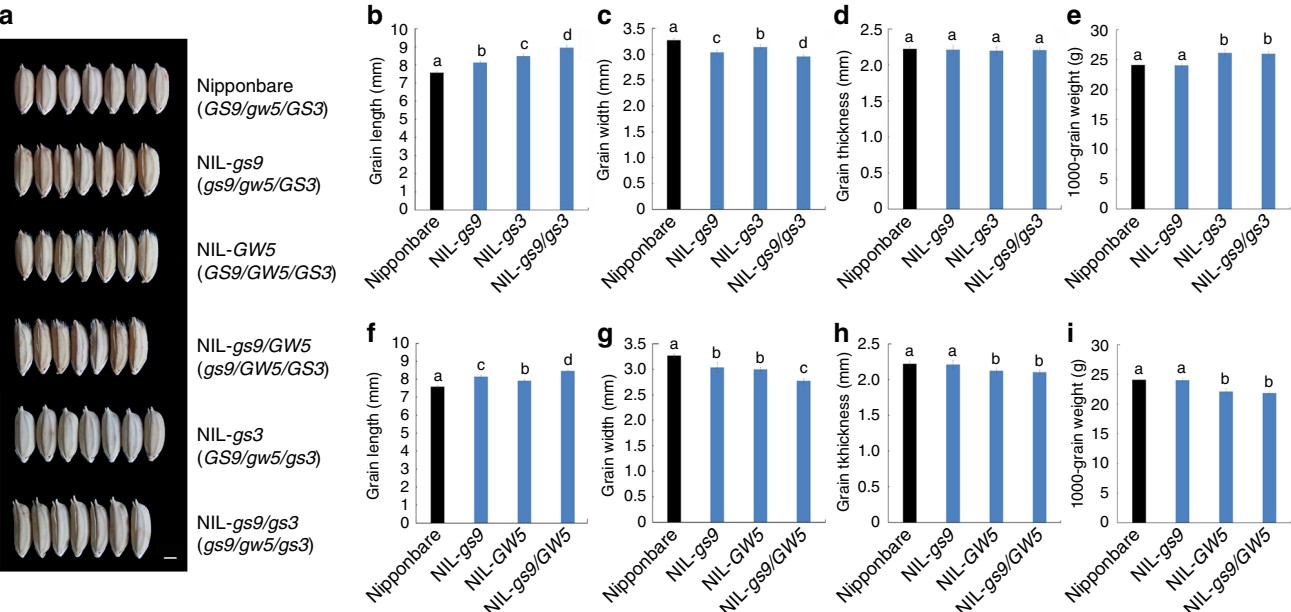

**Fig. 6** Genetic interaction between *GS9* and other grain size genes, *GS3* and *GW5*. **a** Comparison of grain shape among Nipponbare and its various near-isogenic lines (NILs) carrying different combinations of the alleles of *GS9*, *GW3*, and *GW5*. Nipponbare (*GS9/gw5/GS3*) means the wild-type Nipponbare carrying the combination of alleles of *GS9GS9/gw5gw5/GS3GS3*, and the NILs harbored corresponding alleles as shown in Nipponbare background. NIL-*GW5*, near-isogenic line with the *GW5* allele in Nipponbare; NIL-*gs3*, near-isogenic line with the *gs3* allele in Nipponbare; NIL-*gs9/GW5*, pyramiding line from NIL-*gs9* and NIL-*GW5*; NIL-*gs9/gs3*, pyramiding line from NIL-*gs9* and NIL-*gs3*. Scale bar, 2 mm. **b–i** Grain length (**b** and **f**), width (**c** and **g**), thickness (**g** and **h**), and 1000-grain weight (**e** and **i**) of Nipponbare and NILs. Data are given as means ± SD (*n* = 30 in **b–d**, **f**, **g**, *n* = 3 in **e**, **i**)

**GS9 functions independently from other grain size genes**. To further understand the genetic interaction between *GS9* and other known grain size genes, two NILs, NIL-*GW5* and NIL-*gs3*, were constructed in Nipponbare background (Fig. 6a). Subsequently, two pyramiding lines, NIL-*gs9*/*GW5* and NIL-

*gs9*/*gs3*, were generated by crossing with NIL-*gs9*, respectively (Fig. 6a).

The NIL-*GW5* line, which contained the functional *GW5* allele[12] from 9311, showed a slender grain shape (Fig. 6a) but lower grain weight compared with Nipponbare carrying null *gw5*

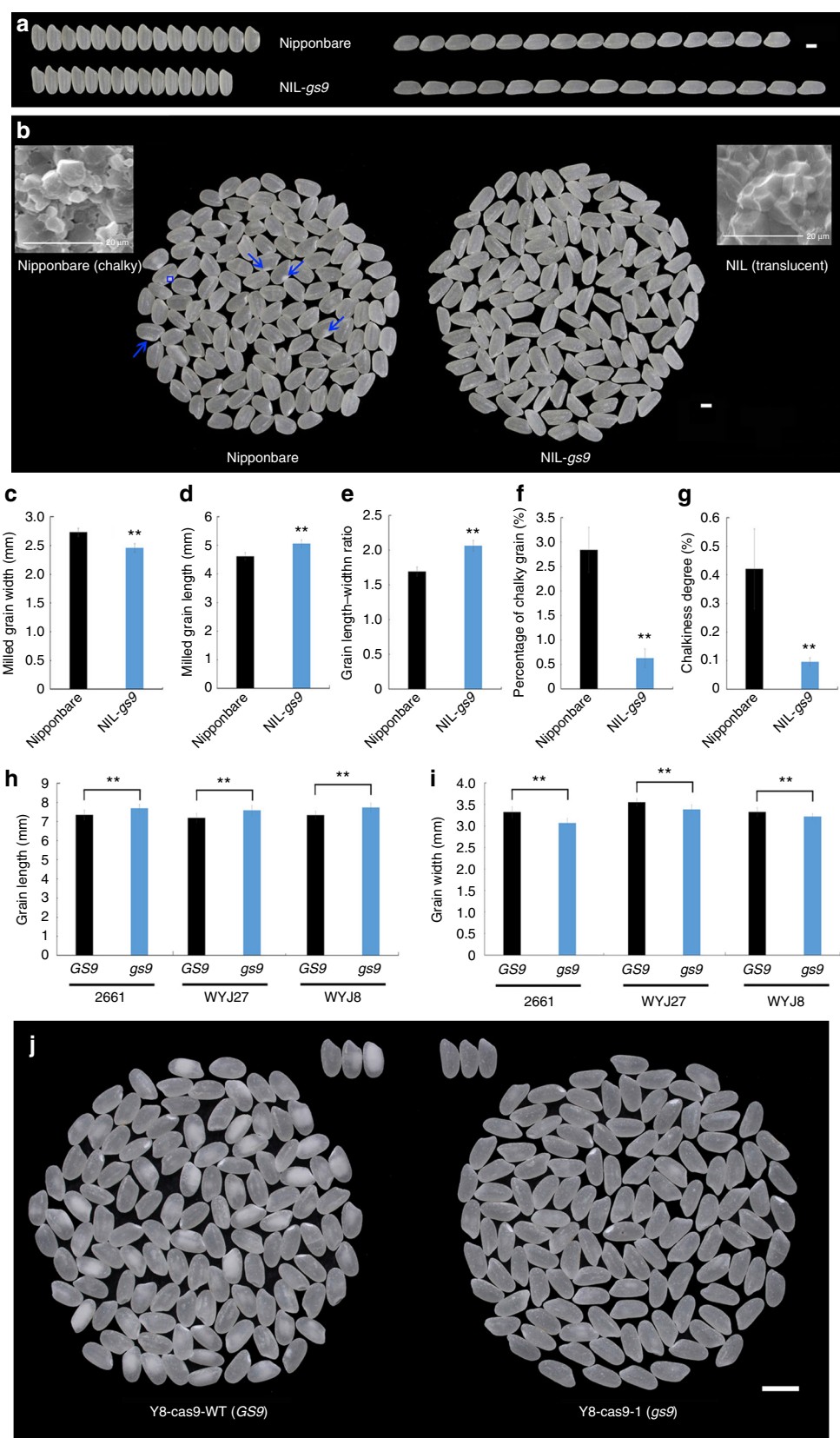

allele (Fig. 6b–e). In the pyramiding line NIL-gs9/GW5, grain shape was much slenderer than that of either NIL-gs9 or NIL-GW5, while grain weight was similar to that of NIL-GW5 (Fig. 6b–e). In addition, GS9 transcript levels were similar between NIL-GW5 and Nipponbare (Supplementary Fig. 20). A similar result was presented between GS3 and GS9 genes. In the NIL-gs3 line carrying null gs3 allele[7] from 9311, the mature grains were much slenderer and heavier than those of Nipponbare (Fig. 6f–i), as previously reported[7]. Meanwhile, in the pyramiding line NIL-gs9/gs3, the grains were much longer and narrower compared with either NIL-gs9 or NIL-gs3, but their weight remained similar to that of NIL-gs3 (Fig. 6f–i). These results suggest that GS9 has an additive effect with GW5 or GS3 in determining grain shape.

The effects of gs9 mutation on the expression of other grain size genes were also examined and no obvious differences were observed between NIL-gs9 and Nipponbare (Supplementary Fig. 21), suggesting no direct correlation between the expression of GS9 and other grain size genes during panicle development in rice.

**Natural variation in the GS9 gene.** To investigate natural variation in GS9 gene, a total of 114 rice germplasms with abundant diversity in grain size were selected. A total of five polymorphic sites were identified (Supplementary Table 3), as described above, between 9311/Qingluzan11 and Nipponbare (Fig. 1h). Thus, a total of five haplotypes, H1–H5, were classified (Supplementary Table 3). The wild rice samples were clustered into four haplotypes excluding H1, with most identified as H2 (Supplementary Table 4), while the cultivar samples were clustered into H1, H2, H4, and H5 haplotypes, with most found in H1 and H5 (Supplementary Table 5). Interestingly, most japonica cultivars belonged to H1 haplotype, while most indica cultivars belonged to H5 (Supplementary Table 3). Phylogenetic tree analysis revealed that the largest genetic distance existed between the H1 and H5 haplotypes (Supplementary Fig. 22).

The two CSSL lines N107 and N139, both carrying the H5 haplotype, had a similar grain shape to Nipponbare, which belonged to the H1 haplotype. These findings imply that natural variation in GS9 gene has no direct correlation with the determination of grain shape. For the rare allele of gs9 from N138, the 7-kb insertion may have occurred spontaneously during CSSL construction.

**Null gs9 allele improves appearance quality of milled rice.** Although grain shape was more slender in NIL-gs9 than Nipponbare, the 1000-grain weight and final grain yield were identical (Fig. 2f, g). Moreover, similar results were presented within the CRISPR/cas9 mutant NPB-cas9-1 (Supplementary Fig. 5e).

Milled rice from NIL-gs9 was more slender than that of Nipponbare (Fig. 7a–e). More importantly, transparency of the milled rice was greatly improved in NIL-gs9. Some milled rice from Nipponbare showed an obvious white core or belly area in the endosperm, unlike NIL-gs9, which showed no or very little chalkiness (Fig. 7b, f, g). In addition, similar to NIL-gs9, milled

rice of NPB-cas9-1 line was also slender without obvious white spots in the endosperm (Supplementary Fig. 5f–i).

The rice quality assay showed a similar brown and milled rice percentage, apparent amylose and protein contents, gel consistency, and gelatinization properties between NIL-gs9 and Nipponbare (Supplementary Fig. 23a–f). Thus, the null gs9 allele appears to greatly improve the appearance quality of milled rice without affecting any other qualities or properties of rice.

**Improve grain appearance quality by introgressing gs9.** As shown above, the null gs9 allele is rare and has great potential to improve grain appearance quality. Two recently released japonica cultivars, Wuyunjing27 (WYJ27) and Wuyunjing8 (WYJ8), with high yield and widely grown in lower reaches of the Yangtze River in China, were therefore selected as targets for introgression of gs9 allele. With the assistance of the gs9-specific molecular marker (Supplementary Fig. 24a), the gs9 allele was successfully introgressed into WYJ8 and WYJ27 via several rounds of backcross, respectively.

As expected, within the BC$_3$F$_2$ population derived from WYJ27 and N138, mature grains of WYJ27-gs9 plants, carrying gs9 allele, were significantly more slender than those of WYJ27-GS9 with normal GS9 allele, with a 5.5% increase in length and 4.7% decrease in width but no changes in thickness or weight (Fig. 7h, i and Supplementary Fig. 24b, c). Similar findings were observed in the WYJ8-gs9 grains, with a 5.4% increase in length and 3.2% decrease in width (Fig. 7h, i and Supplementary Fig. 24b, c). Moreover, introgression of the gs9 allele caused similar improvements in the potential new breeding line 2661, which has a large grain size and high yield. Grains of 2661-gs9 individuals were 4.6% longer and 7.6% narrower than those of 2661-GS9 (Fig. 7h, i and Supplementary Fig. 24b, c).

As a convenient method of utilizing the null gs9 allele, the CRISPR/cas9 editing system was used in Yandao8 (Y8), another high-yield cultivar. One homologous null gs9 mutant, Y8-cas9-1, was identified with a 1-bp insertion (Supplementary Fig. 25a), similar with NPB-cas9-1 mutant in Nipponbare background (Supplementary Fig. 5a). As expected, the grain shape was significantly more slender than that of the non-mutated control (Y8-cas9-WT), with a 5.5% increase in length and 9.1% decrease in width, and no changes in grain thickness (Supplementary Fig. 25b–e). More importantly, the milled rice of Y8-cas9-1 was also much more slender than that of Y8-cas9-WT, resulted in significant decrease of chalkiness (Fig. 7j). These findings confirmed the great potential for application of null gs9 allele to improve grain appearance quality in high-yield but poor appearance rice cultivars.

## Discussion

A number of genes/QTLs are known to control grain size and shape in rice, several of which have been cloned and characterized[6]. However, our understanding remains fragmentary, and thus, further efforts to mine elite genes/alleles are required. It is well-known that grain size is controlled by multiple signaling pathways; however, the gaps in these pathways and their interactions in co-regulating grain size are largely unknown[6, 21].

**Fig. 7** Improved appearance quality of milled rice by introgression of gs9 allele. **a** The shape of milled rice of NIL-gs9 and its recipient Nipponbare. Scale bar, 2 mm. **b** Decreased chalkiness in milled rice of NIL-gs9 compared with those of Nipponbare. Blue arrows indicate chalkiness in Nipponbare. Scale bar, 2 mm. Scanning electron microscopy of cross-sections of milled rice from Nipponbare and NIL-gs9 are shown in the upper left and right, respectively. Scale bar, 20 μm. The length (**c**), width (**d**), length–width ratio (**e**) of whole milled rice from Nipponbare and NIL-gs9. Data are given as means ± SD (n = 30). The percentage of milled rice with chalkiness (**f**) and the chalkiness degree of milled rice (**g**) from Nipponbare and NIL-gs9. Comparison of grain length (**h**) and width (**i**) between the individuals, from the same BC$_3$F$_2$ backcross population but carrying the homozygous GS9 or gs9 alleles, respectively, under japonica 2661, WYJ27, or WYJ8 background. Data are given as means ± SD (n = 30 in **c-e**, n = 3 in **f** and **g**, n = 20 in **h** and **i**); ** significant difference (P < 0.01, t-test). **j** Comparison of the appearance of milled rice between Y8-cas9-WT and Y8-cas9-1, carrying the homozygous GS9 or gs9 alleles, respectively. Scale bar, 5 mm

In the present study, we identified a grain shape regulator, *GS9*, contributing to a slender grain shape (Fig. 1). *GS9* encodes an unknown expressed protein and its homologs are widespread but specific to higher plants, with only one homolog in rice genome (Fig. 4 and Supplementary Fig. 10), indicating the irreplaceable role of *GS9* in grain development and shape determination (Fig. 2 and Supplementary Fig. 7). Overexpression of *GS9* led to changes in the architecture of both vegetative and reproductive organs (Fig. 2 and Supplementary Fig. 6), possibly due to its ectopic expression[7].

*GS9* allelic variation had no effect on the expression of other grain size-related genes (Supplementary Figs. 20 and 21). In addition, genetic interaction using NIL lines revealed that *GS9* is independent of both *GS3* and *GW5*[7, 12, 22] (Fig. 6). Moreover, only full-length GS9 contributes to the regulation of grain shape (Fig. 4 and Supplementary Fig. 13). This is quite different from *GS3*, as several domains of which independently regulate grain size[7]. Taken together, *GS9* functions as a distinct regulator of grain shape, independent of other known grain size genes.

GS9-OsOFP14 interaction was identified and confirmed, resulting in repression of *GS9* transcription activation activity (Fig. 5 and Supplementary Table 2). The OFPs family are transcription repressors, in most cases, functioning in regulation of fruit shape[32]. In *Arabidopsis*, OFPs participate in multiple aspects of plant growth and development[36–38]. The rice genome contains 31 putative *OFP* genes; however, little is known about their functions[33]. Recently, OsOFP8 was found to regulate plant growth via interacting with OsGSK2, a key negative regulator of BR signaling[34]. In this study, GS9 together with OsOFP8 and OsOFP14 form a transcriptional complex (Fig. 5). Correspondingly, the leaves of *GS9*-overexpressing plants were more erect than those of wild-type (Supplementary Fig. 6g), similar to those of *OsOFP8*-RNAi lines[34]. Moreover, increased leaf angle was also observed in NIL-*gs9* plants (Fig. 2a). These findings suggested that *GS9* might play a role in modulating BR pathway, at least partially through its interaction with OsOFP8, thereby regulating rice spikelet development.

The mechanism underlying BR regulation of grain size is quite complex[39], and OFPs are thought to have overlapping and diverse functions in regulating plant growth and development[33, 40]. But the knowledge of their interaction is still limited. OsBZR1 could bind the promoter of *OsOFP1*, and in turn OsOFP1 interacts with DLT and OsGSK2, suggesting that BR may regulates OFP1 at both transcription and protein levels to modulate rice architecture and grain morphology[41, 42]. Moreover, OsOFP19 could also modulate plant architecture and grain shape via integration of cell division and BR signaling[43]. Similarly, GW5 interacts with OsGSK2, repressing its kinase activity and thereby leading to an increase in un-phosphorylated OsBZR1 and DLT, subsequently regulating grain hull size by affecting cell number[25, 39]. OsGSK2 also directly interacts with OsGRF4/GL2, inhibiting its transcription activation activity, and thereby regulating the grain length[15]. Overall, BR also plays an important role in modulating rice grain size by interacting with various regulators. In this study, GS9 together with OsOFP8 and OsOFP14 was found to function as a transcription activity regulator, which may be regulated by OsGSK2 via the bridge of OsOFP8 (Fig. 5). Our results from both *OsOFP8* and *OsGSK2* transgenic rice further validated their function in regulating grain size (Supplementary Figs. 16 and 17). Thus, the molecular mechanism of regulation of grain shape here might rely partially on an OsGSK2-OsOFP8-GS9 signal transduction module, further co-regulating downstream target genes involved in cell division. The transcriptome analysis identified a set of DEGs (Supplementary Data 1), some of which might be the candidate genes involved in determination of grain shape. Thus, it is possible that GS9 forms a linkage between transcription factors and BR signaling pathway during spikelet development; however, details of this potential mechanism require further elucidation.

So far, a number of rice grain size genes, including *GL2*, *GL7*, *GLW7*, and *GSE5*, are known to contribute to the diversity of grain size and shape via expression alteration[10, 15, 20, 44]. Although several SNPs were identified in *GS9*, they did not contribute to the changes in grain shape (Fig. 1h and Supplementary Fig. 1). In contrast, null *gs9* mutation caused a slender grain shape, while *GS9* overexpression resulted in a rounder grain shape (Fig. 2 and Supplementary Fig. 6). These findings suggest that expression levels of *GS9* cause grain shape alterations.

*gs9* null allele caused a slender spikelet hull and cell number was altered in both transverse and longitudinal directions in the spikelet hull of NIL-*gs9* plants (Fig. 3, Supplementary Fig. 7). Therefore, grain shape differences between NIL-*gs9* and wild type could be attributed to cell division in the rice glume. Similar alterations in grain shape and cell proliferation were observed in *GW7*[19]. In general, large spikelet hulls are usually accompanied by incomplete grain filling[14], but slender hulls provide a shorter milk-filling path, and thus, eliminate chalkiness of milled grains of NIL-*gs9* or Y8-cas9-1 line (Fig. 7). Starch granules are loosely packed and spherical in a chalky endosperm[45, 46]. Consistent with this, the starch granules in the chalky area in Nipponbare were spherical and loose, while in NIL-*gs9*, the entire area was translucent (Fig. 7).

Slightly increased leaf angle was observed in NIL-*gs9* plants (Fig. 2a), which may affect the performance of population photosynthesis efficiency, yet it did not alter population yield under conventional management in the field (Fig. 2g). It is possible that the NIL-*gs9* population maintains a suitable level of the field spaced arrangement among individuals to capture sunlight and other resources. In addition, *gs9* null allele showed similar performance to *gw8* and upregulation of *GL7/GW7* in altering grain shape and appearance quality[14, 19, 20]. The effect of *gs9* null allele with slender grains and improved appearance quality were also confirmed using genome editing approach (Supplementary Fig. 5). Overall, the *gs9* null allele seems to be a specific regulator of appearance quality in rice.

Therefore, to mine more *GS9* alleles, we analyzed more than 100 germplasms of both wild and cultivated rice, revealing five haplotypes, but no other loss-of-function allele (Supplementary Tables 5–7). Interestingly, haplotype distribution showed a certain preference among rice subpopulations, and phylogenetic distance among haplotypes well matched the genetic distance between subpopulations (Supplementary Table 3 and Supplementary Fig. 22). However, the common H1 and H5 haplotypes had an equal effect on grain shape (Fig. 1h and Supplementary Fig. 1). This is quite different from the beneficial alleles of *gs3*, *GS5*, *gw5*, and *GW8*, which have been artificially selected in modern rice varieties during domestication[13, 14, 21, 22, 47, 48]. Therefore, it implied that sequence variation in the *GS9* locus had been accrued during rice domestication, and they were naturally reserved due to no or limited effects on plant growth and morphology. Besides, it is also possible that the loss-of-function allele of *GS9* may have been escaped during artificial selection, as the *gs9* mutation lacks an effect on grain weight as well as final yields, and high yielding is highly preferred during modern rice-breeding program.

In conclusion, *gs9* is a rare and specific allele involved in regulating grain shape and appearance quality in rice, and its benefits were shown in several high-yield rice cultivars via introgression of *gs9* allele (Fig. 7 and Supplementary Fig. 24). Moreover, several new high-yield breeding lines with same improvements were also generated using the genome editing approach (Fig. 7 and Supplementary Figs. 5 and 25). Furthermore, QTL pyramiding based on combinations of *gs9* and *GW5/*

*gs3* alleles allowed development of several new rice lines showing improved grain shape (Fig. 6) and possible better grain quality[14]. These findings will help studies aimed at understanding the combined effect of multiple alleles on grain size[49], thereby contributing to improving grain quality of rice.

## Methods

**Plant materials and trait measurements**. The two CSSLs N138 (carrying null *gs9* allele) and N139 (carrying wild-type *GS9* allele) were generated by repetitive backcrossing of progeny derived from a cross between *japonica* rice Nipponbare (recipient) and *indica* rice Qingluzan11 (donor) (Supplementary Fig. 1). Similarly, the CSSL N107 (carrying wild-type *GS9* allele) was generated from repetitive backcross progeny from a cross between Nipponbare (recipient) and *indica* 9311 (donor)[30] (Supplementary Fig. 1).

The near-isogenic line NIL-*gs9*, which possessed the *gs9* allele from the N138 line, was derived from a backcross between progeny of the N138 line, which contains only one donor segment on chromosome 9. During this process, recessive individuals were selected for mapping. Two other NILs, NIL-*gs3* and NIL-*GW5*, were also constructed in the background of Nipponbare, in which null *gs3* and functional *GW5* alleles were introgressed from *indica* 9311, respectively (Fig. 6). The *OsOFP8*-overexpression line (*OsOFP8*-OE7)[34] and the *OsGSK2*-RNAi transgenic line (*OsGSK2*-Gi-2)[35] were derived from the *japonica* cultivar Zhonghua11 (ZH11), which were kindly provided by Prof. Jianxiong Li and Prof. Chengcai Chu, respectively.

A set of germplasms, including 83 rice varieties (*Oryza sativa* L.) and 31 wild rice samples, were used for genotyping of *GS9* (Supplementary Tables 6 and 7). To further analyze the effect of the null *gs9* allele on grain shape, four recently released *japonica* cultivars with high yield but poor grain appearance, Wuyunjing27 (WYJ27), Wuyunjing8 (WYJ8), 2661, and Yandao8 (Y8), were selected for introgression of the *gs9* allele and creation of null *gs9* mutants using the CRISPR/cas9 approach.

Rice plants grew in an experimental field in Yangzhou, China, during the natural growing season. Some transgenic plants grew in greenhouse referring to nature condition. Spikelet hulls were harvested before flowering for measurements of length and width. Grain length, width, and thickness as well as the 1000-grain weight were measured using fully filled grains after maturation. For plot yield tests, all plants in 4-m² plots were collected in three completely randomized blocks. Plant height, flag leaf length and width, and main panicle length were measured on the main culm. In addition, the percentage of chalky grains and degree of chalkiness of milled rice were measured using a grain appearance analyzer ScanMaker (Microtek, China). The apparent amylose content, gel consistency were measured with the standard of NY/T 593-2013 published by Ministry of Agriculture, China (http://www.zbgr.org/27/StandardDetail1476335.htm). Viscosity was tested with a Rapid Visco Analyzer (RVA) (Newport Scientific, Australia) and protein content was estimated by nitrogen combustion with a nitrogen determinator (Kjeltec 8400, FOSS, Denmark)[50].

**Genetic analysis and molecular mapping**. The genetic background of the *gs9* mutant (N138) was confirmed using reduced whole-genome sequencing[30]. A total of 289 $F_2$ plants derived from the cross of N138/Nipponbare were used for genetic analysis and mapping of *GS9* at chromosome 9. An additional 2400 $F_3$ recessive individuals were then used for fine mapping. The phenotype of grain shape in selected recombinants was confirmed using the self-progeny. Several molecular markers were designed using Primer Premier 5.0 based on the genomic sequences of interest between Nipponbare and 9311, and the primer sequences are listed in Supplementary Table 1.

**Microscopy observations**. Fresh young spikelet hulls were fixed, dehydrated, and embedded in Paraplast Plus (Sigma) then cut into 10-μm-thick sections. Cross-sections were analyzed by light microscopy (Olympus), and the cell number and cell area in the outer parenchyma cell layer of hulls were measured using ImageJ and Adobe Photoshop CS2 software. For glume cell observation, the outer surfaces of the spikelet glumes were observed by scanning electron microscope (S-4800, Hitachi). For starch granule observation, natural cross sections of mature milled rice were observed by scanning electron microscope (XL30ESEM, Philip).

**RNA extraction and RT-PCR**. Total RNA was isolated using an RNeasy Plant mini kit (Qiagen) or TRIzol. RNase-free DNase I was used to remove genomic DNA, and subsequently first-strand cDNA was synthesized using reverse transcriptase (Thermo) with the oligo dT18 primer. The full-length or truncated cDNA of *GS9* from Nipponbare was amplified and sequenced, and the rice housekeeping gene, *Actin* (*LOC_Os03g0718100*), used as an internal control. Real-time PCR analysis was performed on ABI ViiA7 according to the SYBR Green method, and the primers were listed in Supplementary Table 6.

**Vector construction and rice transformation**. As a complementation test, a 2283-bp genomic fragment upstream of *GS9/ LOC_Os09g27590* was amplified from

wild-type Nipponbare using primers p1300-590pro (Supplementary Table 6). A 1205-bp cDNA fragment containing the entire coding sequence of *GS9/ LOC_Os09g27590* was also amplified from the cDNAs of Nipponbare using the primers p1300-590FL (Supplementary Table 6) and fused to the upstream promoter region. The fusion fragment was inserted into the binary vector pCAMBIA1300, and the resulting plasmid *pGS9::GS9* was introduced into the callus of NIL-*gs9* by *Agrobacterium*-mediated transformation[51].

The entire coding sequence of *GS9/LOC_Os09g27590* used in complementary analysis was also inserted into the binary vector *pCAMBIA1300* under control of the *CaMV 35S* promoter for overexpression analysis (Supplementary Fig. 6a). The resulting plasmid *p35S::GS9-FL* was introduced into NIL-*gs9* or Nipponbare to generate overexpression lines, respectively.

To genome editing by the CRISPR/Cas9 system[52], a 21-bp target sequence was selected for specific recognition of *GS9/LOC_Os09g27590* (Supplementary Table 6 and Supplementary Fig. 5a). The fragment was cloned into the vector SK-gRNA digested with *Aar* I, then digested and cloned into the *pC1300-Cas9* vector, and the plasmid was transformed into Nipponbare or Yandao8, respectively, to generate the *gs9* mutants.

In addition, different truncations of *GS9/LOC_Os09g27590* based on protein sequence conservation were also amplified and cloned into *pCAMBIA1300* (Supplementary Table 6) under control of the *CaMV 35S* promoter (Supplementary Fig. 13a, b). The resultant plasmids were then transformed into Nipponbare, respectively.

For promoter activity assay, a 2.3-kb *GS9* promoter region from wild-type Nipponbare (Supplementary Table 6) was fused to the *GUS* reporter gene with the nopaline synthase terminator and inserted into the *pCAMBIA1301* vector. The constructed plasmid *pGS9::GUS* was then transformed into Nipponbare for GUS activity by the histochemical assay method.

**Bioinformatic analyses**. Nucleotide and protein sequences of *GS9/ LOC_Os09g27590* based on the full-length cDNA and the Rice Genome Annotation Project (http://rice.plantbiology.msu.edu/) were deposited in the National Center for Biotechnology Information (NCBI) GenBank database (accession number: MF621928). For phylogenetic analysis of *GS9* homologs, a total of 49 protein sequences were obtained through the NCBI BLAST search. Phylogenetic trees were constructed using MEGA 5.0 based on the neighbor-joining method. Eight protein sequences of Poaceae plants were selected for sequence alignment using GeneDoc software. In addition, the coding region of *GS9* was amplified and sequenced in a set of 114 rice germplasms (Supplementary Tables 6 and 7), and the phylogenetic trees of *GS9* haplotypes were constructed using MEGA 5.0 software.

**Transactivation activity assay in yeast**. To construct the necessary serial vectors, full-length coding sequences and their various truncations of *GS9* were amplified and cloned into the *pGBKT7* vector. The truncations of *GS9* for the transactivation activity were the same as those used for rice transformation. The resulting constructs were then transformed into the AH109 yeast strain, and the Matchmaker GAL4-based Two-Hybrid system 3 (Clontech) was used for the transactivation activity assay. Each yeast liquid culture was diluted to an absorbance of 0.5 at OD600, and 3 μl of each dilution inoculated onto tryptophan-, histidine-, and adenine-negative synthetic dropout medium. The relevant PCR primer sequences are given in Supplementary Table 6.

**Y2H screening and confirmation**. Y2H screening was performed using Matchmaker Gold Yeast two-hybrid systems (Clontech). The truncated coding region (GS9-C1, without transcription activation activity) was introduced into the *pGBKT7* vector and the truncated protein used as a bait to screen a cDNA library prepared from developing inflorescence (P1 stage) of Nipponbare. Full-length cDNA of the candidate interacting proteins and several additional proteins including OsOFP1, OsOFP8, and OsGSK2 were amplified and cloned into the pGADT7 vector, respectively, for further verification. In addition, the entire coding regions of *OsOFP14/LOC_Os04g33870* and *OsGSK2* were introduced into the *pGBKT7* vector, respectively. Relevant primers are listed in Supplementary Table 6.

**Transient expression assays**. For subcellular localization analysis, *35S::GS9-GFP* and *35S::OsOFP14-GFP* fusions were constructed by in-frame fusion of full-length *GS9* or *OsOFP14* cDNA with *GFP* (green fluorescent protein) coding region, respectively. The fusion genes were then inserted into the *pCAMBIA2300* or *pJIT163* vectors driven by the *CaMV 35S* promoter, and the constructs were transformed into *Agrobacterium* strain GV3101 and injected into tobacco leaves, or directly transformed into rice protoplasts, respectively. GFP signal was then observed by confocal microscope (Leica). All the primers are listed in Supplementary Table 6.

For BiFC assays, full-length *GS9* or *OsOFP14* cDNA was cloned into pBiFC vectors using the gateway system containing either the N- or C-terminal end of YFP. The resulting constructs (*nYFP-GS9* and *OsOFP14-cCFP*) and corresponding empty vectors were transformed into *Agrobacterium* strain GV3101, respectively. In addition, full-length *GS9*, *OsOFP14*, or *OsOFP8* cDNA was cloned into *pYNE*(R) and *pYCE*(R) containing the N- or C-terminal end of YFP, respectively, and the resulting constructs (*nYFP-GS9*, *cYFP-OsOFP14*, and *cYFP-OsOFP8*) and their

corresponding empty vectors were directly transformed into the protoplasts of rice Nipponbare and *Arabidopsis* Col plants, respectively. Fluorescence was then observed by confocal microscope (Leica)[53]. All the primers are listed in Supplementary Table 6.

**Protoplast transient assays of gene transcription**. To determine overall transcriptional activity of GS9, full-length cDNA of *GS9* or *OsOFP8* was cloned into the pMN6 vector, while that of *OsOFP14*, *OsOFP8*, or *OsGSK2* into the pJIT163 vector without the DBD region, respectively[54]. The resulting vectors, *pMN6-GS9*, *pMN6-OsOFP8*, *pJIT163-OsOFP14*, *pJIT163-OsOFP8*, and *pJIT163-OsGSK2*, were used as effector plasmids, and co-transformed with the reporter plasmid *pGIL* and internal control *pRIL* into protoplasts of *Arabidopsis* Col plants[55]. After incubation at 22 °C for 15 h, the protoplasts were collected for luciferase activity assay (Promega). All the primers are listed in Supplementary Table 6.

**RNA-sequencing analysis**. Young panicles (approximate 5 cm in length) of Nipponbare and NIL-*gs9* plants were harvested for total RNA preparation using TRizol method (Invitrogen), and they were further purified using mRNA purification kit (Promega). Three biological replicates were collected and used for RNA sequencing. The library products were ready for sequencing via Illumina HiSeq 2500. The filtered clean reads were aligned to the rice Nipponbare reference genome and genes (http://rice.plantbiology.msu.edu/) using TopHat2/Bowtie2 software. The Blast2GO (http://www.blast2go.org/) program was used to obtain GO annotations of DEGs and finding significantly enriched GO terms with *P*-value < 0.05.

**Statistical analysis**. Results are presented as mean ± SD. SPSS 16.0 software was used for all statistical tests. Statistical significance was determined by independent-sample *t*-test for comparison of two groups and one-way ANOVA for comparison of three or more groups. Differences were considered statistically significant when $P \leq 0.05$. *P*-values are indicated by * when $P < 0.05$ or ** when $P < 0.01$.

**Data availability**. The data that support the findings of this study are available from the corresponding authors upon request. The sequence of the *GS9* coding region has been deposited in the GenBank nucleotide database under accession code MF621928 https://www.ncbi.nlm.nih.gov/nuccore/MF621928. RNA-sequencing data has been deposited in the SRA database under accession code SRP132484 https://www.ncbi.nlm.nih.gov/sra/?term=SRP132484.

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

## Acknowledgements

This work was supported by the Ministry of Science and Technology of China (2016YFD0100902 and 2016ZX08009003-004) and the National Natural Science Foundation of China (31601275) to Q.-F.L., and China Postdoctoral Science Foundation (2017M610357) to D.S.Z., and Jiangsu PAPD and "333" Projects to Q.Q.L. We thank Dr. Ke-Jian Wang for providing the CRISPR/Cas9 vectors, Prof. Jia-Yang Li for providing the pYNE(R) and pYCE(R) vectors, Prof. Jian-Xiong Li for providing the rice *OsOFP8*-overexpression transgenic rice line, Prof. Cheng-Cai Chu for providing the *OsGSK2*-RNAi transgenic rice line, and Prof. Chen Chen for comments on the article.

## Author contributions

Q.Q.L. and D.S.Z. conceived and designed the experiments. D.S.Z., Q.F.L., C.Q.Z, C.Z., Q.Q.Y., X.Y.R., J.L., and L.X.P. performed the experiments. D.S.Z., Q.F.L., Q.Q.L., and M. H.G. analyzed and interpreted the data. D.S.Z., Q.Q.L., and Q.F.L. wrote the paper. All authors reviewed the manuscript.

## Additional information

**Competing interests:** The authors declare no competing financial interests.

