## [Peer Review File(PDF 120 kb) · Nature Communications]

Reviewers' comments:

Reviewer #1 (Remarks to the Author):

This study identified a novel grain shape gene GS9 that encodes a novel protein. The *gs9* null mutant produced slender grains with better grain transparent appearance and less chalkiness but without affect grain weight. GS9 interacts directly with OsOFP14 and OsOFP8 (the rice ovate family proteins), and functions as a transcriptional activator. GS9 functions independently from other grain size genes, including GS3 and GW5. Introduction of the *gs9* allele into various elite rice cultivars by using either backcrossing or genome editing improved grain shape without interfering with other grain size genes. These results suggest that the GS9 gene may have potential application to optimize grain shape and increase appearance quality in rice breeding. However, although the results show that GS9 interact with OsOFP14 and OsOFP8, the target gene(s) regulated by GS9 and the regulatory pathway are unclear. A transcriptome analysis between WT and *gs9* lines may provide some candidate target genes. In addition, the *gs9* mutant line produced large leaf angle (Figure 2a); this altered plant architecture may affect the performance of photosynthesis efficiency and the yield of the mutant plant population. Another question is that except for the null-mutant allele occurred during the development of the SSSL N138, why function-defective *gs9* alleles are not present in natural rice cultivars? Is this gene important for fitness?

Please correct “grain weigth” to “grain weight” in Figure 2f.

Reviewer #2 (Remarks to the Author):

The present study by Zhao et al. reported the identification of a novel gene, named GS9, controlling grain shape in rice. The study highlighted the function of GS9 in regulating the length/width ratio of grains without affecting other agronomic traits including grain yield. Although the loss-of-function allele was actually derived from a natural mutation but not existed among the natural variations, the functional specificity of the gene render it practical value as a target for improvement of the grain appearance quality, an important trait for marketing consideration. Apparently, the major flaw of the study is about mechanism work regarding the functional relevance between GS9 and its interacting proteins OFPs, which lacks the basic genetic evidence as well as additional supporting information as detailed below.

1. The signal indicating the interactions in BiFC analysis looks weak and the method is prone to bring false positive result, especially in tobacco, a heterogeneous system. Additional verification should be required.
2. It should be detailed whether OFP8 or OFP14 also regulates grain shape. I noted that a recent report showed OFP1 is also involved in regulating grain shape (Front. Plant Sci. 2017, 8: 1698), which may be referred for the authors' consideration or discussion.
3. When the authors tested the interactions among GS9, GSK2, OFP1 and OFP14 and claimed that GS9 or OFP14 cannot interact with GSK2, a positive control confirming the interaction between GSK2 and OFP8 should be included to make sure the vector expressing GSK2 is alright.
4. The authors speculated that GS9 majorly regulates cell division on the longitudinal direction. If this is true, it should be clarified whether OFPs, or GSK2, or BR also regulate grain length by majorly regulating cell division.
5. I suggested the authors to perform the subcellular localization analysis of both GS9 and OFP14 in rice cells, but not only in tobacco leaves, as the two systems sometimes led to different results.

Reviewer #3 (Remarks to the Author):

In this study, the authors identified novel grain shape regulating gene, GS9 by map-based cloning. The authors found that the NIL-gs9 shows increased cell number in longitudinal direction. The authors showed that the GS9 protein functions to a novel transcriptional activator and it interact with OsOFP14 and OsOFP8 in Yeast. The authors also analyzed the epistasis between GS9 and other grain size genes. However, GS9 had an additive effect with GW5 and GS3 in determining grain size and shape. Although the cloning of GS9 gene is solid and reliable, molecular mechanism of grain shape regulation is insufficient.

Major comments:

1. Although the authors showed that the GS9 protein interacts with OsOFP14 and OsOFP8 in yeast, no evidence of this interaction in vivo. The authors should confirm this.
2. Although the authors guessed that the GS9, OsOFP14 and OsOFP8 are involved in transcription co-regulation and this activity seems to be modulated by OsGSK2 kinase, there are

no evidences in this manuscript.

Response to reviewers' comments

Reviewer #1:

This study identified a novel grain shape gene GS9 that encodes a novel protein. The gs9 null mutant produced slender grains with better grain transparent appearance and less chalkiness but without affect grain weight. GS9 interacts directly with OsOFP14 and OsOFP8 (the rice ovate family proteins), and functions as a transcriptional activator. GS9 functions independently from other grain size genes, including GS3 and GW5. Introduction of the gs9 allele into various elite rice cultivars by using either backcrossing or genome editing improved grain shape without interfering with other grain size genes. These results suggest that the GS9 gene may have potential application to optimize grain shape and increase appearance quality in rice breeding.

Response: Thank you for the positive comments.

Comment: *However, although the results show that GS9 interact with OsOFP14 and OsOFP8, the target gene(s) regulated by GS9 and the regulatory pathway are unclear. A transcriptome analysis between WT and gs9 lines may provide some candidate target genes.*

Response: Thanks for your comment and suggestion. Following your suggestion, a transcriptome analysis was performed to explore the functional mechanism of GS9 in the revised manuscript. The new data were shown in Supplementary Tables 3 and 4, and Supplementary Figure 18, and new paragraphs were added to interpret the transcriptome data in Lines 298-317 (Results) and Lines 505-507 (Discussion) of text revised. Although the transcript abundance of genes related to OFPs and BR-signaling showed no remarkable difference between wild-type and NIL-gs9 line (Supplementary Table 4), a number of target genes with distinct transcriptional expression changes (fold change >2 or <0.5) were identified, which were summarized in Supplementary Table 3 and Supplementary Figure 18. We believe that the data

from the transcriptome analysis would help us to further dissect the GS9-responsive molecular regulatory network in regulating grain shape of rice. Thank you again for the valuable advice.

Comment: *In addition, the gs9 mutant line produced large leaf angle (Figure 2a); this altered plant architecture may affect the performance of photosynthesis efficiency and the yield of the mutant plant population.*

Response: Thanks for your comment and suggestion. As you mentioned, the plant architecture is a key parameter to determine rice yield. In the dense planting conditions, rice plants with erect leaf phenotype, for example, mild BR-deficient or BR-insensitive mutant, will have enhanced per-unit area grain yield (Sakamoto et al., 2005). Rice with a certain increment of BR biosynthesis or signaling will led to the changes of rice architecture, including increased leaf angle. In conventional growth conditions, the per-plant grain yield of the BR-enhanced rice will increase (Wu et al., 2008). The data from our yield experiment exhibit no difference between the *gs9* line and its wild-type control (Figure 2g). There might be two possible reasons to this result. Firstly, the increment of the leaf angle is mild in *gs9* mutant. Second, under our conventional planting conditions, the increased leaf angle in *gs9* mutant line might not be serious enough to affect grain yield. The field space arrangement and the population to capture light and other resources still maintain a suitable level in our experiment. Thus, we have revised the discussion in the revised text as followings (Lines 537-541): “In present study, although slightly increased leaf angle was observed in NIL-*gs9* plants (Fig. 2a), which may affect the performance of population photosynthesis efficiency, it did not altered population yield under conventional management in the field (Fig. 2g). It is possible that the NIL-*gs9* population maintains a suitable level of the field spaced arrangement among individuals to capture sunlight and other resources.”

Comment: *Another question is that except for the null-mutant allele occurred during the development of the SSSL N138, why function-defective gs9 alleles are not present*

in natural rice cultivars? Is this gene important for fitness?

Response: Thank you for your critical comments. In our study, a total of 114 germplasms, including 83 rice cultivars and 31 wild rice samples, were used for genotyping of *GS9* locus. No function-defective *gs9* allele was identified in these rice germplasms. However, this cannot completely eliminate the possibility that the function-defective *gs9* exists in the untested rice germplasms. Another possibility is that due to a yield preference of early human selection, function-defective *gs9* with slender grain shape but lack of yield contribution may have escaped human selection. Due to the lack of selection pressure, the naturally reserved sequence variation of *GS9* gene showed no consistent phenotype alterations. We have revised the description in the revised Discussion section (Lines 561-566) as followings: “...Therefore, it implied that sequence variation in the *GS9* locus had been accrued during rice domestication, and they were naturally reserved due to no or limited effects on plant growth as well as morphology. Besides, it is also possible that the loss-of-function allele of *GS9* may have been escaped during artificial selection, as the *gs9* mutation lacks an effect on grain weight as well as final yields, and high yielding is highly preferred during modern rice breeding program.”

Comment: Please correct “grain weighth” to “grain weight” in Figure 2f.

Response: Thanks very much and we are sorry for this mistake. We have corrected this in Figure 2f of revised manuscript.

Reviewer #2:

*The present study by Zhao et al. reported the identification of a novel gene, named *GS9*, controlling grain shape in rice. The study highlighted the function of *GS9* in regulating the length/width ratio of grains without affecting other agronomic traits*

including grain yield. Although the loss-of-function allele was actually derived from a natural mutation but not existed among the natural variations, the functional specificity of the gene render it practical value as a target for improvement of the grain appearance quality, an important trait for marketing consideration. Apparently, the major flaw of the study is about mechanism work regarding the functional relevance between GS9 and its interacting proteins OFPs, which lacks the basic genetic evidence as well as additional supporting information as detailed below.

Response: Thank you for the comments and suggestions. Based on your suggestions, we have carried out several additional experiments and got more solid data to support the interaction between GS9 and its interacting proteins. (1) An additional BiFC analysis in rice and *Arabidopsis* protoplast system was applied to further confirm the GS9-OsOFP8 and GS9-OsOFP14 interactions *in vivo* (Figure 5b, Supplementary Figure 14). Moreover, the interaction between OsGSK2 and OsOFP8 was further confirmed by Y2H analysis (Figure 5f). (2) We further tested the interplays among GS9, OsOFP14, OsOFP8 and OsGSK2 in co-regulating GS9 transcriptional activity by using dual-luciferase assays system. The results confirmed the repression effects of OFPs on GS9 transcriptional activity, and this repression effect could be partially recovered by OsGSK2 (Figure 5c,d). (3) More importantly, we got the mature seeds of *OsOFP8* and *OsGSK2* transgenic lines from previous reports of Prof. Jianxiong Li (Referance#34) and Prof. Chengcai Chu (Referance#35). The results showed obviously that overexpression of *OsOFP8* (Supplementary Figure 16) or down-regulation of *OsGSK2* (Supplementary Figure 17) could generate the similar longer grain morphology by affecting cell division as *gs9* mutant. It provided solid data to support the genetic interaction between GS9 and identified proteins. The detailed information about these revisions could be referred in the following responses.

Comment 1. *The signal indicating the interactions in BiFC analysis looks weak and the method is prone to bring false positive result, especially in tobacco, a heterogeneous system. Additional verification should be required.*

Response: Thanks for your suggestion. According to your suggestions, we have repeated the BiFC analysis in tobacco cells and verified the interaction between GS9 and OsOFP14 as showed in Supplementary Figure 14a. Besides, we further performed the BiFC analysis in rice and *Arabidopsis* protoplasts, which confirmed the interaction as shown in revised Figure 5b and Supplementary Figure 14. We have added related descriptions in the Results section (Lines 256-259) of revised text. Therefore, we believe that all these evidences support the interaction between GS9 with OsOFP14 *in vivo*.

Comment 2. *It should be detailed whether OFP8 or OFP14 also regulates grain shape. I noted that a recent report showed OFP1 is also involved in regulating grain shape (Front. Plant Sci. 2017, 8: 1698), which may be referred for the authors' consideration or discussion.*

Response: Thank you very much. As you suggested, we got the mature seeds of *OFP8*-overexpression lines from Jianxiong Li's lab as reported (Reference#34), and our data showed that the mature grains also exhibited slender grain shape similar to that of *gs9* mutant (Supplementary Figure 16), which consistent with its role in negatively regulating *GS9* transcription activity. We have added related descriptions in the Results section (Lines 283-296) of revised text as followings: "With the end to clarify whether *OsOFP8* and *OsGSK2* also regulate grain size, we obtained their transgenic rice lines as previous reported^{34, 35}. Interestingly, the *OsOFP8*-overexpression line (*OsOFP8-OE7*)³⁴ exhibited a more slender grain phenotype than that of wild-type *Zhonghua11* (ZH11), with a 7% increase in grain length (Supplementary Fig. 16a-c). ... The results of scanning electron microscopy showed no significant differences in longitudinal cell density on the outer surface of the glume between *OsOFP8-OE7* or ... and their corresponding wild-types (Supplementary Fig. 16d, 17d). Thus, it suggested that the longer grain length phenotype of *OsOFP8*-overexpression or *OsGSK2*-RNAi lines was the result from an increase in longitudinal cell numbers of their spikelet hull. Overall, these results

further indicated that GS9-OsOFP8-OsGSK2 complex participate in regulation of grain shape, at least in part, by modulating cell division.”

Moreover, we thank for your introduction of the recent report on OsOFP1 (*Front. Plant Sci.* 2017, 8: 1698), and we also learned another recent report on OsOFP19 (*Plant J.* 2017, 10.1111/tpj.13793). Both OFPs were reported to be involved in regulating grain shape in rice, though with different extent of effects. In higher plants, OFPs were firstly reported to determine the transition from round to pear-shaped fruit in tomato. All these data suggest that rice OFPs might involve in regulating grain shape in different participatory approaches. We have discussed these in the revised text (Lines 485-492) as followings: “More recently, Xiao et al reported that OsBZR1 could bind the promoter of OsOFP1 gene, and in turn OsOFP1 protein interacts with DLT and OsGSK2, suggesting that BR may regulates OFP1 at both transcription and protein levels to modulate plant architecture and grain morphology in rice⁴¹. OsBZR1 is the critical transcription factor of BR signaling⁴², and DLT acts as the direct target of OsGSK2 to positively regulate BR signaling³⁵. Moreover, OsOFP19 also could modulate plant architecture and grain shape via integration of cell division and BR signaling, by formation of a functional complex among OsOFP19, DLT and OSH1⁴³.”

Comment 3. *When the authors tested the interactions among GS9, GSK2, OFP1 and OFP14 and claimed that GS9 or OFP14 cannot interact with GSK2, a positive control confirming the interaction between GSK2 and OFP8 should be included to make sure the vector expressing GSK2 is alright.*

Response: Thanks so much for your suggestion. According to your suggestion, we have performed an additional experiment and added the data on the interaction between OsGSK2 and OsOFP8 as a positive control, which was added in Figure 5f of the revised manuscript. Therefore, our results further confirmed that OsGSK2 indeed interacts with OsOFP8, as reported by Yang et al. (2016), but not with OsOFP14 and GS9 (Figure 5).

Comment 4. *The authors speculated that GS9 majorly regulates cell division on the*

longitudinal direction. If this is true, it should be clarified whether OFPs, or GSK2, or BR also regulate grain length by majorly regulating cell division.

Response: Thank you very much. As suggested, fortunately, we got the mature seeds of *OsOFP8* and *OsGSK2* transgenic lines from previous reports of Prof. Jianxiong Li (Reference#34) and Prof. Chengcai Chu (Reference#35). Our results showed that overexpression of *OsOFP8* (Supplementary Figure 16) or down-regulation of *OsGSK2* (Supplementary Figure 17) could generate similar longer grain morphology by affecting cell division as *gs9* mutant. We had added related descriptions in the Results section (Lines 283-296) of revised text as followings: “With the end to clarify whether *OsOFP8* and *OsGSK2* also regulate grain size, we obtained their transgenic rice lines as previous reported^{34,35}. Interestingly, the *OsOFP8*-overexpression line (*OsOFP8-OE7*)³⁴ exhibited a more slender grain phenotype than that of wild-type *Zhonghua11* (ZH11), with a 7% increase in grain length (Supplementary Fig. 16a-c). In *OsGSK2-Gi-2* transgenic line³⁵, suppression of *OsGSK2* expression by RNA interference (RNAi) resulted in a 22% increase in grain length compared with that of the wild-type ZH11 (Supplementary Fig. 17a-c). The results of scanning electron microscopy showed no significant differences in longitudinal cell density on the outer surface of the glume between *OsOFP8-OE7* or *OsGSK2-Gi-2* and their corresponding wild-types (Supplementary Fig. 16d,17d). Thus, it suggested that the longer grain length phenotype of *OsOFP8*-overexpression or *OsGSK2*-RNAi lines was the result from an increase in longitudinal cell numbers of their spikelet hull. Overall, these results further indicated that *GS9-OsOFP8-OsGSK2* complex participate in regulation of grain shape, at least in part, by modulating cell division.”

Comment 5. *I suggested the authors to perform the subcellular localization analysis of both GS9 and OFP14 in rice cells, but not only in tobacco leaves, as the two systems sometimes led to different results.*

Response: Thanks so much. According to your suggestion, we have performed the subcellular localization analyses of both GS9 and OsOFP14 in rice protoplast cells.

The result showed that both GS9 and OsOFP14 localized in the nucleus of rice cells (Figure 4c), which consistent with that from the tobacco cell system. We have added the new result in the revised manuscript (Line 223).

Reviewer #3:

In this study, the authors identified novel grain shape regulating gene, GS9 by map-based cloning. The authors found that the NIL-gs9 shows increased cell number in longitudinal direction. The authors showed that the GS9 protein functions to a novel transcriptional activator and it interact with OsOFP14 and OsOFP8 in Yeast. The authors also analyzed the epistasis between GS9 and other grain size genes. However, GS9 had an additive effect with GW5 and GS3 in determining grain size and shape. Although the cloning of GS9 gene is solid and reliable, molecular mechanism of grain shape regulation is insufficient.

Response: Thank you for your comments. By referring to all editorial and reviewers' suggestions, we have designed and performed additional experiments to facilitate the further exploration of the molecular mechanism of grain shape regulation. A number of new data were integrated into the revised manuscript. For example, we further confirmed the protein-protein interaction by performing BiFC in the rice and Arabidopsis protoplast system. Also, OsOFP14, OsOFP8 and OsGSK2 involved co-regulation of the transcriptional activity of GS9 was also dissected. Furthermore, we demonstrated that GS9, OsOFP8 and OsGSK2 are all involved in the regulation of grain length by modulating cell division in the longitudinal direction of the rice glume. Finally, RNA-sequencing data provided some candidate downstream target genes that might mediate GS9-regulated grain development.

Major comment 1. *Although the authors showed that the GS9 protein interacts with*

OsOFP14 and OsOFP8 in yeast, no evidence of this interaction in vivo. The authors should confirm this.

Response: Thanks for the suggestion. In our previous version of manuscript, we confirmed the interaction between GS9 and OFP14 in both yeast and tobacco epidermal cells. In the revised manuscript, we first repeated the interaction between GS9 and OsOFP14 in tobacco leaf cells, and then we further verified the GS9-OFP14 and GS9-OFP8 interactions in rice and Arabidopsis protoplast by using BiFC analysis. The result is consistent with that observed in the tobacco system, hence confirming that GS9 could interact with both OsOFP14 and OsOFP8 in the nucleus of plant cells. The new results (Figure 5b and Supplementary Figure 14) and the interpretation of the results (Lines 256-259) were added in the revised manuscript.

Major comment 2. *Although the authors guessed that the GS9, OsOFP14 and OsOFP8 are involved in transcription co-regulation and this activity seems to be modulated by OsGSK2 kinase, there are no evidences in this manuscript.*

Response: Thank you for your comments and suggestions. According to your suggestions, we have carried out more experiments and tested again the hypothesis by using the pMN6 transient transcriptional activity assay system. Our data showed that both OsOFP14 and OsOFP8 can suppress GS9 transcriptional activity, while OsGSK2 alleviates the suppression effects likely by directly interacting with OFPs (Figure 5d). And we have revised the text (Lines 275-279) as followings: “The dual-luciferase assays also revealed the repression effect of OsOFP8 on the transcriptional activity of GS9 (Fig. 5c,d), and this repression could be partly recovered by OsGSK2 (Fig. 5d and Supplementary Fig. 15). As expected, co-expression of OsOFP14 and OsOFP8 had a more serious repression effect on GS9 transcriptional activity, and it could also be attenuated by OsGSK2 (Fig. 5d).”. Overall, these findings suggest that OsGSK2 kinase are involved in the regulation of grain shape in rice partially by modulating the transcription activities of OFPs and GS9 directly or indirectly.

Reviewers' Comments:

Reviewer #1 (Remarks to the Author):

This revision has been greatly improved by adding some experiments, which meet the comments and suggestions of the reviewers.

Reviewer #2 (Remarks to the Author):

I have no further suggestions since the authors have addressed my previous concerns.

Reviewer #3 (Remarks to the Author):

In the revised manuscript, the authors have addressed or answered all comments and questions raised by the first peer review. The manuscript is greatly improved and I have no more comment.

Response to reviewers' comments

Reviewer #1 (Remarks to the Author):

This revision has been greatly improved by adding some experiments, which meet the comments and suggestions of the reviewers.

Response: Thank you for the positive comment.

Reviewer #2 (Remarks to the Author):

I have no further suggestions since the authors have addressed my previous concerns.

Response: Thank you for the positive comment.

Reviewer #3 (Remarks to the Author):

In the revised manuscript, the authors have addressed or answered all comments and questions raised by the first peer review. The manuscript is greatly improved and I have no more comment.

Response: Thank you for the positive comment.